# GENERATIVE IMPUTATION AND STOCHASTIC PREDICTION

## ABSTRACT

In many machine learning applications, we are faced with incomplete datasets. In the literature, missing data imputation techniques have been mostly concerned with filling missing values. However, the existence of missing values is synonymous with uncertainties not only over the distribution of missing values but also over target class assignments that require careful consideration. In this paper, we propose a simple and effective method for imputing missing features and estimating the distribution of target assignments given incomplete data. In order to make imputations, we train a simple and effective generator network to generate imputations that a discriminator network is tasked to distinguish. Following this, a predictor network is trained using the imputed samples from the generator network to capture the classification uncertainties and make predictions accordingly. The proposed method is evaluated on CIFAR-10 image dataset as well as three real-world tabular classification datasets, under different missingness rates and structures. Our experimental results show the effectiveness of the proposed method in generating imputations as well as providing estimates for the class uncertainties in a classification task when faced with missing values.

## 1 INTRODUCTION

While a large body of the machine learning literature is built upon the assumption of having access to complete datasets, in many real-world problems only incomplete datasets are available. The existence of missing values can be due to many different causes such as human subjects not adhering to certain questions or features not being collected frequently due to financial or experimental limitations, sensors failures, and so forth. Data imputation techniques have been suggested as a solution to bridge this gap in the literature by replacing missing values with observed values.

Missing data imputation approaches can be categorized into single and multiple imputation methods. Single imputation methods try to replace each missing value with a plausible value that is the best fit given the value of other correlated features and knowledge extracted from the dataset (Hastie et al., 1999; Anderson, 1957). While these methods are easy to implement and use in practice, imputed values may induce bias by eliminating less likely but important values. Also, these methods do not suggest a way to measure to what extent the imputed values are representative of the missing values (Little & Rubin, 2019).

Multiple imputation (MI) techniques, as suggested by the name, try to use multiple imputed values to impute each missing value. The result would be having a set of imputed datasets that enables measuring how consistent and statistically significant are the results of the experiments (Rubin, 1976). While MI offers interesting statistical insights about the reliability of analysis on incomplete data, the insight is imprecise as it is mainly concerned about the population of data samples rather than individual instances. Specifically, MI methods reason about the statistical properties on a limited number of imputed datasets (less than 10 in most practical implementations) on the population of samples within the dataset (Schafer & Graham, 2002; Murray et al., 2018).

The existence of missing values is synonymous with having uncertainty over these values that requires careful consideration. In many real-world applications, we are dealing with supervised problems that demand modeling and prediction based on incomplete data. Take for instance, prediction of class assignments given an image in which a large portion of the frame is missing. In such a scenario, based on observed frame parts, there might be multiple probable class assignments each having a different likelihood. Here, we are not only interested in imputing missing values or measuring how

robust our imputations are, but also it is highly desirable to measure the impact of missing values on the prediction outcome for each instance.

In this paper, we propose the idea of Generative Imputation and Stochastic Prediction (GI) as a novel approach to impute missing values and to measure class uncertainties arising from the distribution of missing values. The suggested approach is based on neural networks trained using an adversarial objective function. Additionally, a predictor is trained on the generated samples from the imputer network which is able to reflect the impact of uncertainties over missing values. This enables measuring different prediction outcomes and certainties for each specific instance. We evaluate the effectiveness of the proposed method on different incomplete image and tabular datasets under various missingness structures. [1]

## 2   RELATED WORK

One of the simplest traditional methods for handling missing values includes imputing the occurrences of missing values with constant values such as zeros or using mean values. To enhance the accuracy of such imputations, alternatives such as k-nearest neighbors (KNN) (Hastie et al., 1999) and maximum likelihood estimation (MLE) (Anderson, 1957) have been suggested to estimate values to be used given an observed context. While these methods are easy to implement and analyze, they often fail to capture the complex feature dependencies as well as structures present in many problems.

Rubin (1976) suggested a categorization for missingness mechanisms: missing completely at random (MCAR), missing at random (MAR), and missing not at random (MNAR). Under the assumption of MAR, the authors suggested multiple imputation (MI) as a stochastic imputation method. Here, instead of imputing missing values using a single value, several values are sampled to represent the distribution over the missing value. MI generates a few imputed complete datasets that are then used independently in statistical modeling (Schafer & Graham, 2002; Little & Rubin, 2019). Usually, the final goal of MI is to measure the robustness of the final statistical analysis amongst the imputed datasets. In other words, it measures the quality of imputations and the statistical significance of analysis on the imputed data. It should be noted that the number of imputations used in MI is usually very limited. Also, often strong simplifying assumptions are made in modeling the data distribution (e.g., multi-variate Gaussian or Student's t distribution) which limit the applicability of this method (Schafer & Graham, 2002; Murray et al., 2018).

More recently, autoencoder architectures have been suggested as powerful density estimators capable of capturing complex distributions. Perhaps, denoising autoencoders (DAE) (Vincent et al., 2008) are one of the most intuitive approaches in which a neural network is trained to reconstruct and denoise its input. Following a more probabilistic perspective, variational autoencoders (VAE) (Kingma & Welling, 2013) try to learn the data generating distribution via a latent representation. Specifically, conditional variational autoencoders (CVAE) (Sohn et al., 2015) can be used to sample missing values conditioned on observed values. For instance, Mattei & Frellsen (2018) suggested a method based on deep latent variable models and importance sampling that offers a tighter likelihood bound compared to the standard VAE bound. While these methods are powerful generative models applicable to missing data imputation, often samples generated using autoencoders are biased toward the mode of the distribution (e.g., resulting in blurry images, for vision tasks) (Goodfellow et al., 2014; Dumoulin et al., 2016).

Recently, due to the success of generative adversarial networks (GAN), there has been great attention toward applying them to impute missing values. For instance, Yoon et al. (2018) suggested an imputation method based on adversarial and reconstruction loss terms. Li et al. (2019) introduced the idea of using separate generator and discriminator networks to learn the missing data structure and data distribution. These methods have been quite successful and are able to present the state-of-the-art results. Though it should be noted that often the presence of additional loss terms may bias the generated samples toward the mode of the distribution being modeled. Also, these methods are often complicated to be applied in practical setups by practitioners. For instance, Yoon et al. (2018) requires setting hyperparameters to adjust the influence of an MSE loss term as well as the rate of

---

[1]We plan to include a link to the source code and GitHub page related to this paper in the camera-ready version.

Figure 1: Block diagram of the proposed adversarial imputation method. $h$ represents the blending function of (1), and $L$ is the adversarial loss function of (2).

discriminator hint vectors. Also as another example, Li et al. (2019) uses three generators and three discriminators for the final imputer architecture.

From the perspective of supervised analysis, imputation and handling missing values are usually considered as a preprocessing step. A few exceptions exist such as Bayesian models and decision trees that permit direct analysis on incomplete data (Nielsen & Jensen, 2009; Zhang et al., 2005). Note that while certain Bayesian methods such as probabilistic Bayesian networks allow handling of missing values as unobserved variables. However, given an incomplete training dataset and without any known causal structure as a priori, learning such models is a very challenging problem with the complexity of at least NP-complete to learn the network architecture in addition to an iterative EM optimization to learn model parameters (Darwiche, 2009; Neapolitan et al., 2004). We argue that the simplistic approach of imputing missing values as a preprocessing step discards uncertainties that exist in original incomplete data samples. Instead, there is a need for methods that reflect these uncertainties on the final predicted target distribution. This work suggests the idea of training a predictor on different imputed samples to capture the uncertainties over class assignments. Compared to MI, the suggested method interleaves imputation and training a downstream prediction model, enabling to estimate classification uncertainties for each instance.

## 3 PROPOSED METHOD

### 3.1 PROBLEM DEFINITION

In this paper, we make the general assumption of having access to an incomplete dataset $\mathcal{D}$ consisting of a set of feature vector, mask vector, and target class pairs $(\boldsymbol{x}_i, \boldsymbol{k}_i, y_i)$. For each feature vector, $\boldsymbol{x}_i \in \mathbb{R}^d$, only a subset of the features is available. The mask vector $\boldsymbol{k}_i \in \{0, 1\}^d$ is used to indicate available features and missing features by ones and zeros, respectively. Here, to represent features as fixed-width vectors, arbitrary (or NaN) values are used to fill missing values. Also, for convenience, we often use $\boldsymbol{x}_i^{obs}$ and $\boldsymbol{x}_i^{miss}$ to refer to the set of observed and missing features for the feature vector $\boldsymbol{x}_i$.

We define our objective in two steps: $(\boldsymbol{i})$ Imputing missing values via sampling from the conditional distribution of missing features given observed features i.e., $P(\boldsymbol{x}_i^{miss} | \boldsymbol{x}_i^{obs})$. $(\boldsymbol{ii})$ Estimating the distribution of target classes given the observed features and the distribution of missing features i.e., $P(y|\boldsymbol{x}_i^{obs}, \boldsymbol{x}_i^{miss})$. For the first part, we are interested in sampling from the conditional distribution rather than finding the mode of the distribution as the most probable imputation. Similarly, for the second part, we are interested in obtaining a distribution over the possible target assignments and the confidence of each class rather than maximum likelihood class assignments.

### 3.2 GENERATIVE IMPUTATION

To generate samples from the distribution of missing features conditioned on the observed features, we follow the idea first suggested by Yoon et al. (2018). In this paradigm, a generator network is responsible for generating imputations while a discriminator is trying to distinguish imputed features from observed features (see Figure 1).

Specifically, the generator function $G(\boldsymbol{x}_i, \boldsymbol{k}_i, \boldsymbol{z}) \in \mathbb{R}^d$ generates an imputed feature vector, based on observed features, the corresponding mask, and a Gaussian noise vector ($\boldsymbol{z}$). Note that, in order to achieve the final imputed vector, $\widehat{\boldsymbol{x}}_i$, we blend (or, merge) the output of the generator with the input

features to replace generated values with the exact values of observed features:

$$\widehat{\boldsymbol{x}}_{i,j} = \begin{cases} \boldsymbol{x}_{i,j} & \text{if } \boldsymbol{k}_{i,j} = 1 \\ G(\boldsymbol{x}_i, \boldsymbol{k}_i, \boldsymbol{z})_j & \text{if } \boldsymbol{k}_{i,j} = 0 \end{cases}, \tag{1}$$

where $\boldsymbol{x}_{i,j}$ refers to $j$'th feature of sample $i$. Also, note that by sampling $\boldsymbol{z}$ multiple times, we can obtain different imputation samples from the conditional distribution indicated by $\widehat{\boldsymbol{x}}_i^l$ where $l$ is the sample number.

A discriminator network, $D(\widehat{\boldsymbol{x}}_i)$, is trained to distinguish real and imputed features by generating a predicted softmax mask output, $\widehat{\boldsymbol{k}}_i$. Here a binary cross-entropy loss per mask element is used as the adversarial objective function:

$$\max_G \min_D L(G, D) = \mathbb{E}_{\boldsymbol{k} \sim \mathcal{D}, \widehat{\boldsymbol{k}} \sim D(G(\boldsymbol{x}, \boldsymbol{k}, \boldsymbol{z}))} [\boldsymbol{k}^T \log(\widehat{\boldsymbol{k}}) + (1 - \boldsymbol{k})^T \log(1 - \widehat{\boldsymbol{k}})]. \tag{2}$$

The intuition behind this adversarial loss function is that given a generator function which captures the data distribution successfully, the discriminator would not be able to distinguish the parts of the feature vector that were originally missing.

Compared to Yoon et al. (2018), the objective function of (2) does not have an MSE loss term. Instead, we use recent advances in GAN stabilization and training to improve the training process (see Section 3.4). While it is quite prevalent in the adversarial learning literature to use additional loss terms such as mean squared error (MSE) to enhance the quality of generated samples, we decided to keep our solution as simple as possible. Additionally, in our experiments, we provide supporting evidence that this simple loss function enables us to sample from the conditional distribution and prevents biased inclinations toward distribution modes.

### 3.3 STOCHASTIC PREDICTION

To capture the distribution of target classes given incomplete data, we suggest the idea of stochastic prediction. As indicated in the previous section, the generator can be used to sample from the conditional distribution. Here, a predictor is trained based on the imputed samples to predict class assignments and to calculate the confidence of these assignments. For instance, for a specific test sample at hand, if a certain missing feature is a strong indicator of the target class, we would like to observe the impact of different imputations for that feature on the final hypothesis.

Formally, we are interested in finding the certainty of class assignments given observed features:

$$\Psi = P(y|\boldsymbol{x}_i^{obs}). \tag{3}$$

Here, $\Psi$ is a vector where each element is representing a certain class. Rewriting (3) as a marginal we have:

$$\Psi = \int P(\boldsymbol{x}_i^{miss}) P(y|\boldsymbol{x}_i^{obs}, \boldsymbol{x}_i^{miss}) \, d\boldsymbol{x}_i^{miss}. \tag{4}$$

Approximating the integration using a summation, given enough samples, $\Psi$ can be estimated by:

$$\Psi \approx \frac{1}{N} \sum P(y|\boldsymbol{x}_i^{obs}, \widehat{\boldsymbol{x}}_i^{miss}), \tag{5}$$

where $\widehat{\boldsymbol{x}}_i^{miss}$ are samples taken from the conditional distribution of missing features given observed ones. We use the suggested generative imputation method to generate samples required for this approximation. Rewriting (1) using Hadamard product and as function of the noise vector:

$$\widehat{\mathbf{x}}_i = \boldsymbol{k}_i \odot \boldsymbol{x}_i + (1 - \boldsymbol{k}_i) \odot G(\boldsymbol{x}_i, \boldsymbol{k}_i, \boldsymbol{z}) \tag{6}$$

Assuming that a predictor, $F_\theta$, is available which predicts class assignments for a complete feature vector, $\Psi$ can be estimated as:

$$\Psi = \mathbb{E}_{\mathbf{z}}[F_\theta(\widehat{\mathbf{x}}_i)] \approx \frac{1}{N} \sum_{l=1}^N F_\theta(\widehat{\boldsymbol{x}}_i^l) . \tag{7}$$

Algorithm 1 presents the suggested algorithm for training the predictor. It consists of taking samples from the incomplete dataset, then imputing them using our generator network, and using the imputed

**Algorithm 1:** Training the predictor.

**Input:** $G$ (trained imputer), $\mathcal{D}$ (dataset)
**Output:** $F_\theta$ (trained predictor)
**foreach** *Training Epoch* **do**
    **foreach** $(\boldsymbol{x}_i, \boldsymbol{k}_i, y_i)$ *in* $\mathcal{D}$ **do**
        $\boldsymbol{z} \sim N(0, I)$
        $\widehat{\mathbf{x}}_i \leftarrow \boldsymbol{k}_i \odot \boldsymbol{x}_i + (1 - \boldsymbol{k}_i) \odot G(\boldsymbol{x}_i, \boldsymbol{k}_i, \boldsymbol{z})$
        $y_i^{pred} \leftarrow F_\theta(\widehat{\mathbf{x}}_i)$
        $loss \leftarrow L(y_i, y_i^{pred})$
        Backpropagate $loss$
        Update $F_\theta$

**Algorithm 2:** Estimating target distributions.

**Input:** $F_\theta$ (trained predictor), $(\boldsymbol{x}, \boldsymbol{k})$ (test sample), N (ensemble samples)
**Output:** $\Psi$ (distribution over target classes)
$\Psi \leftarrow zeros \in R^{\#classes}$
**foreach** *Ensemble Sample 1 to N* **do**
    $\boldsymbol{z} \sim N(0, I)$
    $\widehat{\mathbf{x}} \leftarrow \boldsymbol{k} \odot \boldsymbol{x} + (1 - \boldsymbol{k}) \odot G(\boldsymbol{x}, \boldsymbol{k}, \boldsymbol{z})$
    $y^{pred} \leftarrow F_\theta(\widehat{\mathbf{x}})$
    $j \leftarrow argmax(y^{pred})$
    $\Psi_j \leftarrow \Psi_j + \frac{1}{N}$

samples to update the predictor. Note that, on each epoch and for each sample, the generator generates a new sample from the conditional distribution. Intuitively, it means that the predictor observes and learns to operate under different imputations for a given sample. This is different from approaches such as multiple imputation where several predictors are trained on different imputed versions of a dataset.

Algorithm 2 presents the suggested algorithm for making predictions and estimating target distributions given a trained predictor model. Here, a sample is imputed $N$ times and inference on this set results in an ensemble of predictions over different imputations. The output of this algorithm can be interpreted as a distribution over the confidence of class assignments given a partially observed test sample. The following claims justify the validity of Algorithm 1 and Algorithm 2.

**Claim 1.** *(Generalization of the predictor). If we assume imputed $\widehat{x}_i$s are samples from the underlying feature distribution, then the assigned training set labels can be modeled as labels generated from a noisy labeling process.*

Claim 1 permits the analysis of the generalization and convergence for the predictor trained using Algorithm 1 based on current literature in training models with noisy labels (Natarajan et al., 2013; Reed et al., 2014; Chen et al., 2019). From the analysis provided by Chen et al. (2019), test accuracy in asymmetric label noise conditions is a quadratic function of the label noise:

$$P(y_i = \widehat{y}_i) = (1 - \epsilon)^2 + \epsilon^2, \tag{8}$$

where $\widehat{y}_i$ is underlying true label for the imputed feature vector ($\widehat{x}_i$), and $y_i$ is the label provided by the incomplete dataset. In (8), label noise ratio, $\epsilon$, represents the probability of the label transition from a certain target class to another:

$$\epsilon = 1 - P(\widehat{y}_i = j | y_i = j). \tag{9}$$

In practice, $\epsilon$ is determined by the problem-specific underlying data distribution as well as the distribution of missing values.

Justification for claim 1 is straightforward, assume that $\{\widehat{y}_i^1 \dots \widehat{y}_i^N\}$ are underlying true labels for each of $\{\widehat{x}_i^1 \dots \widehat{x}_i^N\}$. During training, for any imputed sample in $\{\widehat{x}_i^1 \dots \widehat{x}_i^N\}$, we use the dataset provided label, $y$, to calculate the loss and to update model parameters. In the case that any of $\{\widehat{y}_i^1 \dots \widehat{y}_i^N\}$ is different from $y$, the loss term corresponding to that term would be calculated using a wrong label. Here, if we consider the average impact on gradients for batches of samples rather than individual cases, the overall impact on training would be very similar to the case of training using noisy labels.

**Claim 2.** *(Approximation of the target distribution). If we assume:*
*(i) imputed $\widehat{x}_i$s are valid samples from the underlying feature distribution,*
*(ii) a good predictor can be trained using the incomplete data,*
*(iii) enough samples are used and the Monte Carlo estimator is unbiased,*
*then the target distribution, $\Psi$, can be estimated accurately.*

This claim supports Algorithm 2 that is suggested to estimate the target distribution given a partially observed feature vector.

The first assumption is consistent with the theoretical analysis of generative adversarial networks that they can converge to the true underlying distribution (Arora et al., 2018; Liu et al., 2017). The second assumption is supported by Claim 1. Regarding the last assumption, each sample requires one forward computation of the generator and predictor networks which, based on the scalability of current network architectures, usually permits thousands of samples to be taken at a reasonable computational cost.

### 3.4 IMPLEMENTATION DETAILS

As we conduct experiments on image and tabular datasets, we use different architectures for each. For image datasets, we used a generator and discriminator architectures similar to the ones suggested by Wang et al. (2018). However, we improved these architectures using self-attention layers (Zhang et al., 2018). It should be noted that, while Zhang et al. (2018) suggests using a single self-attention layer in the middle of the network, we observed consistent improvements by inserting multiple self-attention layers before each residual block within the network. Furthermore, as input to the generator, we concatenate input image, mask, and a random $z$ frame along the channels dimension and use it as input. For tabular datasets, we use a simple 4 layer network consisting of fully-connected and batch-norm layers. Also, the input to the generator is the concatenation of a feature vector, mask vector, and a $z$ vector of size $\frac{1}{8}$ of the input. For all experiments, we use an ensemble size ($N$) equal to 128.

We used Adam (Kingma & Ba, 2014) for model optimization. Two time-scale update rule (TTUR) (Heusel et al., 2017) was used to balance training the generator and discriminator networks. We explored best TTUR learning-rate settings from the set of {0.001, 0.0005, 0.0001, 0.00005}. Here, Adam parameters $\beta_1$ and $\beta_2$ are set to 0.5 and 0.999, respectively. Also, spectral normalization was used to stabilize both the generator and discriminator network in our experiments with image data (Miyato et al., 2018). For the predictor network, we used the default Adam settings as suggested by Kingma & Ba (2014). In all training procedures, we decay learning rate by a factor of 5 after reaching a plateau. For all experiments, we use a batch size of 64. Based on our experiments, we found that pretraining the discriminator while fixing the generator network for the first 5% of the training epochs helps the stability of training.

Further detail on exact architectures, experiments, software dependencies, etc. as well as ablation studies is provided in the appendices.

## 4 EXPERIMENTS

### 4.1 DATASETS

To evaluate the proposed method we use CIFAR-10 (Krizhevsky & Hinton, 2009) as an image classification dataset as well as three non-image datasets: UCI Landsat (Dua & Graff, 2017)[2], MIT-BIH arrhythmia (Moody & Mark, 2001), and Diabetes classification (Kachuee et al., 2019) [3]. CIFAR-10 dataset consists of 60,000 32x32 images from 10 different classes. For this task, we use train and test sets as provided by the dataset. As a preprocessing step, we normalize pixel values to the range of [0,1] and subtract the mean image. The only data augmentation we use for this task is to randomly flip training images for each batch.

UCI Landsat consists of 6435 samples of 36 features from 6 different categories. We follow the same train and test split as provided by the dataset. MIT-BIH dataset consists of annotated heartbeat signals from which we used the preprocessed version available online[4] consisting of 92062 samples of 5 different arrhythmia classes. Diabetes dataset is a real-world health dataset of 92,062 samples and 45 features from different categories such as questionnaire, demographics, medical examination, and lab results. The objective is to classify between three different diabetes conditions i.e., normal, pre-diabetes, and diabetes. As MIT-BIH and Diabetes datasets do not provide explicit train and test sets, we randomly select 80% of samples as a training set and the rest as a test set. To preprocess our tabular datasets, statistical and unity based normalization are used to balance the variance of

---

[2]`https://archive.ics.uci.edu/ml/datasets/Statlog+(Landsat+Satellite)`
[3]`https://github.com/mkachuee/Opportunistic`
[4]`https://www.kaggle.com/shayanfazeli/heartbeat`

different features and center them around zero. Also, while different encoding and representation methods are suggested in the literature to handle categorical features (Jang et al., 2016; Nazabal et al., 2018), in this paper, we take the simple approach of encoding categorical variables using one-hot representation and smoothing them by adding Gaussian noise with zero mean and variance equal to 5% of feature variances. In our experiments, we observed a reasonable performance using the suggested simple smoothing; however, more advanced encoding methods are also applicable in this setup and can be applied to enhance the performances even further.

## 4.2 MISSINGNESS MECHANISMS

In our experiments, we consider MCAR uniform and MCAR rectangular missingness structures. In MCAR uniform, each feature of each sample is missing based on a Bernoulli distribution with a certain missingness probability (i.e., missing rate) independent of other features. In addition to the case of uniform missingness, for image tasks, we use rectangular missingness/observation structure where rectangular regions of dataset images are missing/observed. To control the rate of missingness and decide on the regions that are missing for each case, we use a latent beta distribution that samples rectangular region's width and height such that the average missing rate is maintained. For missing rates less than 50% we make the assumption of having a random rectangular region to be missing, whereas for missing rates more than 50% we assume that only a random rectangular region is observed and the rest of the image is missing.

We would like to note that while the suggested solution in this paper is readily compatible with MAR structures, in our experiments, to simplify the presentation of results and to have a fair comparison with other work that does not support the MAR assumption, we limited the scope of our experiments to MCAR. Furthermore, to simulate incomplete datasets and to make sure the same features are missing without explicitly storing masks, we use hashed feature vectors to seed random number generators used to sample missing features. More detail is provided in Appendix C.

## 4.3 EVALUATION MEASURES

Fréchet inception distance (FID) (Heusel et al., 2017) score is used to measure the quality of missing data imputation in experiments with images[5]. We also considered using root means squared error (RMSE); however, we decided not to use this measure as we observed an inconsistent behavior using RMSE in our comparisons as RMSE favors methods that show less variance rather than realistic and sharp samples from the distribution. Also, for each dataset and each missingness scenario, we report top-1 classification accuracy based on the majority vote estimated using Algorithm 2. Another measure that we use in this paper is the comparison between the estimated target certainties and average accuracies achieved for each confidence assignment. We run each experiment multiple times: 4 times for CIFAR-10 and 8 times for tabular datasets. We report the mean and standard deviation of results for each case.

We compare our results with MisGAN (Li et al., 2019) and GAIN (Yoon et al., 2018) as the state of the art imputation algorithms based on GANs as well as basic denoising autoencoder (DAE) (Vincent et al., 2008) and multiple imputation by chained equations (MICE) (Buuren & Groothuis-Oudshoorn, 2010) as baselines. For experiments using MisGAN, we used the same architectures and hyper-parameters as suggested by the MisGAN authors[6]. The only modification was to adapt the last generator layer to generate images with resolutions as we use. Regarding GAIN, we used the same network architecture as our implementation of GI and hyper-parameters as used by the GAIN authors[7]. In the DAE implementation, due to the incomplete data assumption, only observed features appear in the loss function, ignoring reconstruction terms corresponding to missing features. Due to scalability issues, we were only able to use MICE for the smaller non-image datasets. For these methods, to train and evaluate classifiers, we use predictors trained on imputed datasets rather than the stochastic predictor suggested in Algorithm 1.

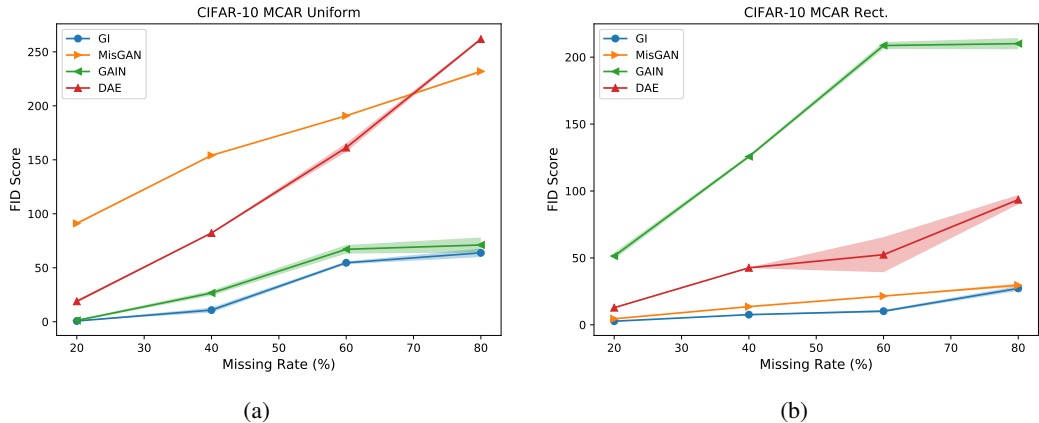

Figure 2: Comparison of FID scores on CIFAR-10 dataset for (a) uniform and (b) rectangular missingness. Lower FID score is better. In many cases, variance values are very small and only observable by magnifying the figures.

## 4.4 RESULTS

Figure 2 presents the comparison of FID scores on the CIFAR-10 dataset at different missing rates for uniform and rectangular missingness. As it can be inferred from these plots, GI outperforms other alternatives in all cases. Also, it can be seen that GAIN is able to provide more reasonable results for uniform missing data structure compared to MisGAN which is mainly effective in the rectangular missing data structure. One possible explanation for this behavior might be the fact that GAIN has an MSE loss term acting similar to an autoencoder loss smoothing noisy missing pixels. On the other hand, MisGAN tries to explicitly model missingness structure and is more successful in capturing a more structured missingness such as the case of a rectangular structure. Table 1 provides a comparison between the top-1 classification accuracy achieved using each method at different missing rates and structures. From this table, GI outperforms other work by achieving the best results in 5 out of 6 cases[8].

Table 2 presents a comparison of classification accuracies for Landsat, MIT-BIH, and Diabetes datasets at different missing rates. In the Landsat benchmark, GI outperforms other work in all cases. Regarding the MIT-BIH experiemts, GI outperforms other work for missing rates more than 30% while achieving similar accuracies to GAIN for lower missing rates. In the diabetes classification task, GI appears to be most effective imputing missing rates more than 20%.

Figure 3 shows a comparison of accuracy versus certainty plots for GI, MisGAN, and GAIN on Landsat dataset at the missing rate of 40%. To generate these figures we trained each imputation method and then used Algorithm 1 to train predictors on imputed samples. Finally, Algorithm 2 used to measure the average accuracy at different prediction confidence levels based on a sample of 128 imputations for each test example. As it can be seen from the plots, GI provides results closest to the ideal case of having average confidence values equal to average accuracies.

## 4.5 VISUALIZATION USING SYNTHESIZED DATA

In order to provide further insight into the operation of GI and how imputations can potentially influence the outcomes of predictions, we conduct experiments on a synthesized dataset. The original underlying data distribution is generated by sampling 5000 samples from 4 Gaussians of standard deviation 0.1 centered on the vertices of a unit square. We assign two different classes to each cluster

---

[5]`https://github.com/mseitzer/pytorch-fid` is adapted to measure the FID scores.

[6]`https://github.com/steveli/misgan`

[7]`https://github.com/jsyoon0823/GAIN`

[8]An earlier version of this paper reported results that are different from the current manuscript. The current version is using the stochastic predictor exclusively on the suggested imputation method and trained using more precise hyper-parameter settings.

Table 1: Top-1 CIFAR-10 classification accuracy for different missing rates and structures.

| | Accuracy at Missing Rate (%) | | | | | |
| | MCAR Uniform | | | MCAR Rect. | | |
| Method | 20% | 40% | 60% | 20% | 40% | 60% |
|---|---|---|---|---|---|---|
| GI | **89.5** (±0.45) | **87.1** (±0.54) | 80.3 (±0.26) | **84.0** (±0.03) | **76.9** (±0.03) | **66.1** (±0.16) |
| MisGAN | 86.5 (±0.31) | 83.7 (±0.40) | 78.7 (±0.26) | 82.9 (±0.44) | 75.6 (±0.20) | 65.0 (±0.31) |
| GAIN | 88.7 (±0.45) | 86.0 (±0.86) | **81.8** (±0.03) | 81.7 (±0.03) | 73.6 (±0.35) | 58.4 (±1.66) |
| DAE | 88.0 (±0.22) | 84.0 (±0.50) | 79.8 (±0.71) | 83.3 (±0.64) | 75.5 (±0.44) | 63.8 (±0.24) |
| Mean | 85.7 (±0.02) | 83.4 (±0.38) | 79.2 (±0.16) | 82.7 (±0.15) | 75.3 (±0.16) | 64.0 (±0.32) |

Table 2: Comparison of classification accuracies for Landsat, MIT-BIH, and Diabetes datasets at different missing rates.

| Dataset | Method | Accuracy at Missing Rate (%)[a] | | | |
| | | 10% | 20% | 30% | 40% |
|---|---|---|---|---|---|
| **Landsat** (Dua & Graff, 2017) | GI | **89.9** (±0.36) | **89.6** (±0.36) | **89.0** (±0.03) | **88.0** (±0.22) |
| | MisGAN | 87.2 (±0.01) | 85.7 (±0.19) | 84.0 (±0.61) | 82.9 (±0.75) |
| | GAIN | 89.7 (±0.42) | 89.4 (±0.56) | 88.4 (±0.71) | 87.7 (±0.10) |
| | DAE | 89.4 (±0.10) | 88.6 (±0.54) | 87.5 (±0.14) | 86.6 (±0.21) |
| | MICE | 89.5 (±0.16) | 89.3 (±0.10) | 88.1 (±0.49) | 87.5 (±0.03) |
| **MIT-BIH** (Moody & Mark, 2001) | GI | **98.5** (±0.02) | **98.4** (±0.03) | **98.2** (±0.07) | **97.7** (±0.03) |
| | MisGAN | 97.8 (±0.13) | 97.4 (±0.07) | 96.7 (±0.07) | 96.2 (±0.09) |
| | GAIN | **98.5** (±0.02) | **98.4** (±0.06) | 98.0 (±0.09) | 97.5 (±0.18) |
| | DAE | 98.4 (±0.02) | 98.2 (±0.11) | 97.9 (±0.09) | 97.4 (±0.02) |
| | MICE | 98.4 (±0.01) | 98.3 (±0.01) | 98.1 (±0.01) | 97.5 (±0.12) |
| **Diabetes** (Kachuee et al., 2019) | GI | 89.6 (±0.13) | **89.0** (±0.03) | **88.2** (±0.62) | **86.8** (±0.38) |
| | MisGAN | 89.7 (±0.01) | 88.9 (±0.30) | 87.6 (±0.02) | 86.4 (±0.68) |
| | GAIN | 89.2 (±0.09) | 88.3 (±0.02) | 86.9 (±0.09) | 83.8 (±1.44) |
| | DAE | 89.3 (±0.05) | 88.2 (±0.19) | 86.9 (±0.09) | 84.8 (±0.03) |
| | MICE | **89.8** (±0.08) | 88.8 (±0.01) | 88.0 (±0.08) | 86.1 (±0.02) |

[a]Baseline accuracies for complete datasets (zero missing rate) are equal to 90.9%, 98.6%, and 90.7% for Landsat, MIT-BIH and Diabetes, respectively.

such that diagonal vertices are of the same class (see Figure 4a, classes are represented with colors). From this underlying distribution, we make an incomplete dataset with 50% of values missing.

The incomplete synthesized dataset is used to train GI and other imputation methods. We take a random test sample in which the second feature has a value of about 0.1 and the other feature is missing. Ideally, in the imputation phase, we would like to sample from the condition distribution i.e. $P(x_1|x_2 = 0.1)$ (see Figure 4b). Here, in the prediction phase, an ideal method would decide on not making a confident classification and report the uncertainty. Note that solely observing the value of 0.1 for the second feature does not provide any useful evidence for the prediction. Figure 4c-f provide samples and classification results for GI, MisGAN, GAIN, and DAE. As it can be inferred from these figures, GI generates reasonable samples from the conditional distribution and also reflects this uncertainty over the prediction. On the other hand MisGAN, probably due to its complexity of using three different generators and discriminator pairs, is suffering from mode collapse and is unable to generate samples from the other class, resulting in over-confident assignments. GAIN, perhaps due to the MSE loss terms, is inclined towards the mean of the conditional distribution at the origin.

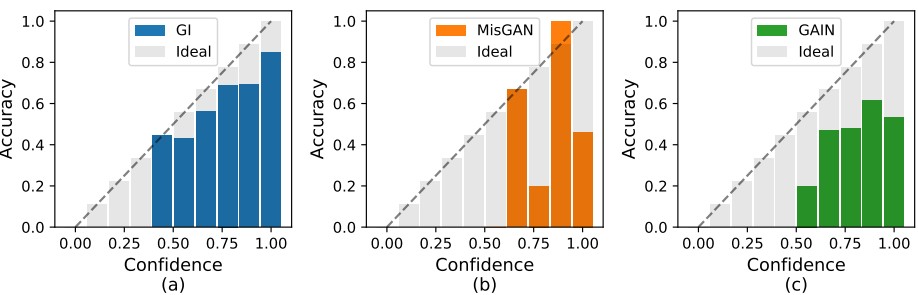

Figure 3: Accuracy versus certainty plots for (a) GI, (b) MisGAN, and (c) GAIN on Landsat dataset at the missing rate of 40%.

DAE, as expected, due to its MSE loss term, only captures the expected value of the distribution mean hence reducing the MSE error and generates over-smoothed imputations.

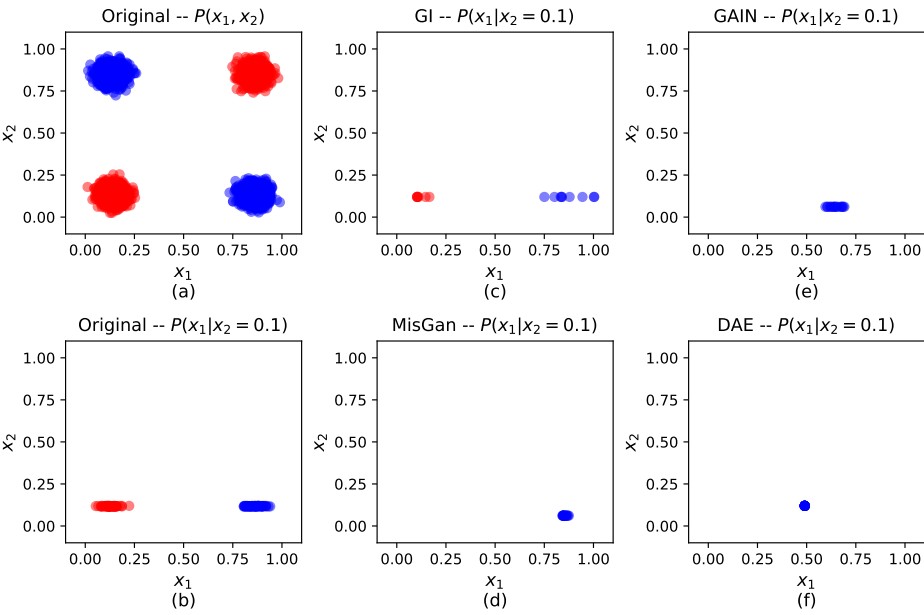

Figure 4: Evaluation using synthesized data: (a) samples from the underlying distribution, (b) samples from the conditional underlying distribution, (c-f) samples from the conditional distribution generate by GI, MisGAN, GAIN, and DAE.

## 5   CONCLUSION

In this paper, we proposed a novel method to generate imputations and measure uncertainties over target class assignments based on incomplete feature vectors. We evaluated the effectiveness of the suggested approach on image and tabular data via using different measures such as FID distance, classification accuracy, and confidence versus accuracy plots. According to the experiments, the proposed method not only can generate accurate imputations but also is able to model prediction uncertainties arising from missing values. The proposed method is applicable to many real-world applications where only an incomplete dataset is available, and modeling classification uncertainties is a necessity.

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

## A    IMPLEMENTATION AND EXPERIMENTS

Table 3 presents the list of software dependencies and versions used in our implementation. To produce results related to this paper, we used a workstation with 4 NVIDIA GeForce RTX-2080Ti GPUs, a 12 core Intel Core i9-7920X processor, and 128 GB memory. Each experiment took between about 4 hours to 48 hours, based on the task and method being tested.

Table 3: Software dependencies.

| Dependency | Version |
|---|---|
| python | 3.7.1 |
| pytorch | 1.0.0 |
| cuda100 | 1.0 |
| ipython | 7.2.0 |
| jupyter | 1.0.0 |
| numpy | 1.15.4 |
| pandas | 0.24.1 |
| scikit-learn | 0.20.1 |
| scipy | 1.1.0 |
| torchvision | 0.2.1 |
| tqdm | 4.28.1 |
| matplotlib | 3.0.1 |

## B    NETWORK ARCHITECTURES

Table 4 shows the exact architectures used in this paper. To show each layer or block we used the following notation. `CxSyPz-t` represents a 2-d convolution layer of kernel size x, stride y, padding z, and number of output channels t followed by ReLU activation. `Attn` represents a self-attention layer similar to Zhang *et al.*[9]. `R-x` represents a residual block consisting of two 2-d convolutions with kernel size 3 (padding size 1), batch normalization, and ReLU activation. `CTxSyPz-t` is the convolution transpose corresponding to `CxSyPz-t`. `FC-x` is representing a linear fully-connected layer of x output neurons with biases. We use spectral normalization as suggested by Miyato *et al.*[10] for all convolutional layers in both generator and discriminator networks.

Table 4: Network architectures used in our experiments.

| Dataset | Generator/Discriminator Architecture | Predictor Architecture |
|---|---|---|
| **CIFAR-10** | C7S1P3-64, C3S2P1-128, Attn, R-128, Attn, R-128, Attn, R-128, Attn, R-128, CT3S2P1-128, CT7S1P3-3, Tanh/Sigmoid | ResNet-18 [11,12] |
| **Landsat** | FC-64, Sigmoid, BNorm, FC-64, Sigmoid, BNorm, FC-64, Sigmoid, BNorm, FC-36, Tanh/Sigmoid | FC-64, ReLU, BNorm, FC-64, ReLU, BNorm, FC-6, Softmax |
| **MIT-BIH** | FC-1860, ReLU, BNorm, FC-1860, ReLU, BNorm, FC-1860, ReLU, BNorm, FC-186, Tanh/Sigmoid | FC-1860, ReLU, BNorm, FC-1860, ReLU, BNorm, FC-5, Softmax |
| **Diabetes** | FC-45, ReLU, BNorm, FC-45, ReLU, BNorm, FC-45, ReLU, BNorm, FC-45, Tanh/Sigmoid | FC-22, ReLU, BNorm, FC-22, ReLU, BNorm, FC-3, Softmax |

---

[9]Zhang, Han, et al. "Self-attention generative adversarial networks." arXiv preprint arXiv:1805.08318 (2018).
[10]Miyato, Takeru, et al. "Spectral normalization for generative adversarial networks." arXiv preprint arXiv:1802.05957 (2018).

## C  MISSING DATA MECHANISMS

In this paper, we conduct experiments on two mechanisms for missing values: MCAR uniform and MCAR rectangular. As in our experiments and comparisons, we consider the case where only an incomplete dataset is available for training. It is crucial to guarantee that each method has only access to a unique incomplete version of each sample. However, it is relatively expensive to load and store feature masks for each sample in the dataset. Instead, we generate missing values during the data load for each batch. A hashing mechanism is used to ensure that the same parts are missing for each sample throughout the training. Note that we set system, python, and external library hash seeds to fixed values to ensure the consistency between different runs.

Algorithm 3 presents the procedure used for generating missing values with uniform structure. This algorithm is sampling independent Bernoulli distributions with probabilities equal to the missing rate. Algorithm 4 shows the outline for the rectangular missing structure used in image experiments. It consists of selecting a random point as the center of the rectangle and then deciding on parameters to be used for the beta distribution based on the missing rate. Finally, the width and height of the rectangular region are sampled from the latent beta distribution. In other words, we generate rectangular regions centered at random locations within the image which have width and height values determined by samples from a latent beta distribution. Here, distribution parameters, $\alpha$ and $\beta$, are used to control the average missing rate. The outcome would be rectangular regions of different shape at different locations within the frame with the expected portion of missing area equal to the missing rate.

In order to decide on the beta distribution parameters i.e. $\alpha$ and $\beta$ we use numerical simulations. Specifically, we fix one of the parameters to 1 and change the other parameter in the range of [1,10], while measuring the average missing rate caused by each case. Figure 5 shows the missing rates caused by different beta distribution parameters. The first half of Figure 5 (missing rates less than about 0.18) corresponds to setting $\beta$ to 1 and changing $\alpha$ values; and the other half fixing $\alpha$ to 1 and changing $\beta$ values. To generate missing rates more than 50% we invert our masks and limit the observation to the rectangular region while the rest of the image is missing. Note that missing rates indicate the ratio of features that are missing on the average case. As we are using a latent model for sampling width and height for the rectangles, the actual missing ratios for each specific sample differs between samples. See Table 5 for visual examples of different missing rates and missing structures.

---

**Algorithm 3:** MCAR uniform generation.

---

**Input:** $\boldsymbol{x}$ (complete feature), $r$ (missing rate)
**Output:** $\boldsymbol{x_m}$ (incomplete feature)
$seed_x \leftarrow hash(\boldsymbol{x})$
$\boldsymbol{k} \leftarrow 1 - Bernoulli(seed_x, shape(x), prob = r)$
$\boldsymbol{x_m} \leftarrow \boldsymbol{k} \odot \boldsymbol{x} + (1 - \boldsymbol{k}) \odot NaN$

---

**Algorithm 4:** MCAR rect. generation.

---

**Input:** $\boldsymbol{x}$ (complete feature), $r$ (missing rate)
**Output:** $\boldsymbol{x_m}$ (incomplete feature)
$seed_x \leftarrow hash(\boldsymbol{x})$
$n_x, n_y \leftarrow shape(\boldsymbol{x})$
$(p_x, p_y) \sim (uniform(0, n_x), uniform(0, n_y))$
$\alpha, \beta \leftarrow beta\_params(r)$ // $beta\_params$ gives $\alpha, \beta$ for
    each missing rate based on numerical
    simulations
$(w, h) \sim (Beta(\alpha, \beta) \times n_x), Beta(\alpha, \beta) \times n_y))$
$\boldsymbol{k} \leftarrow rect\_mask(p_x, p_y, w, h)$
$\boldsymbol{x_m} \leftarrow \boldsymbol{k} \odot \boldsymbol{x} + (1 - \boldsymbol{k}) \odot NaN$

---

[11]He, Kaiming, et al. "Deep residual learning for image recognition." Proceedings of the IEEE conference on computer vision and pattern recognition. 2016.
[12]https://github.com/kuangliu/pytorch-cifar

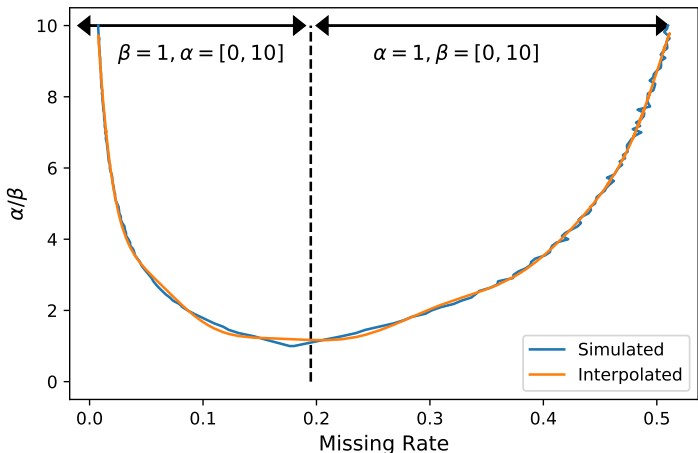

Figure 5: Simulation results for measuring average missing rate given different beta distribution parameters.

Table 5: Examples of uniform and rectangular missing structures at different missing rates.

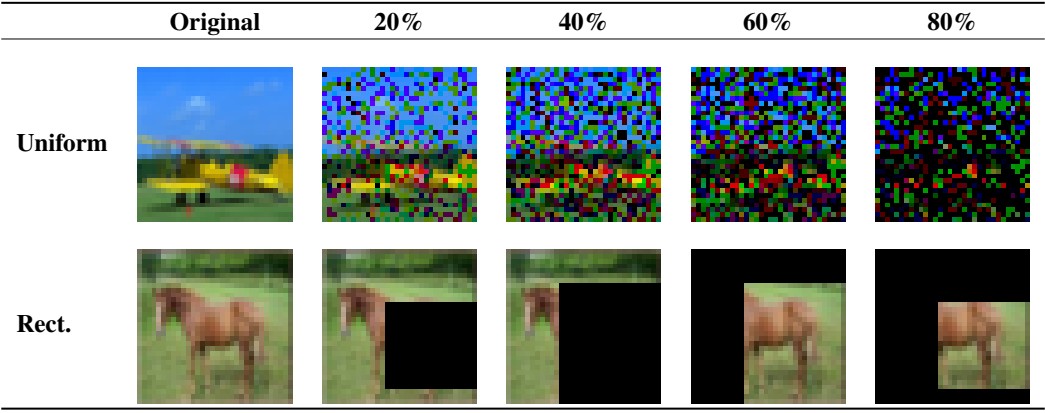

## D ABLATION STUDY

Figure 6 presents a comparison between using (GI W/ Atten.) and not using (GI W/O Atten.) self-attention layers before each residual block in the proposed architecture. We report FID scores on CIFAR-10 with rectangular missingness. As it can be inferred from this comparison, using self-attention achieves a consistent improvement over the baseline. We also examined the case of uniform missingness; however, we did not observe any significant improvement for this case. One possible explanation could be the fact that imputing missing data with a uniform structure can be done by processing local regions and does not require attending to different distant regions across the image.

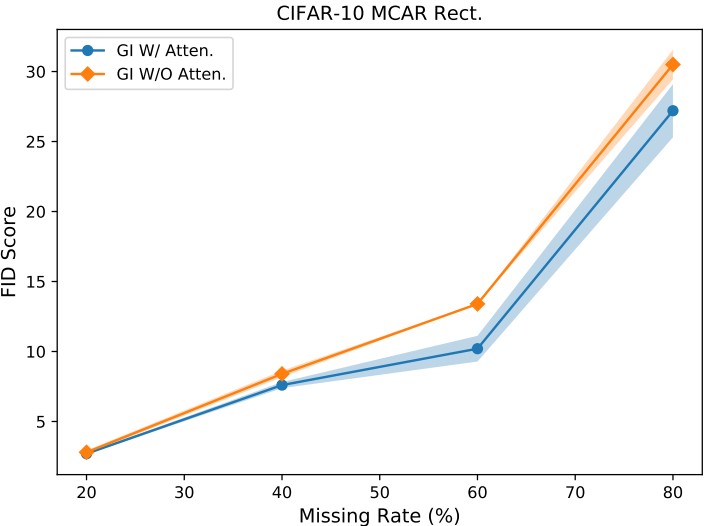

Figure 6: Comparison of FID scores achieved with (GI W/ Atten.) and without (GI W/O Atten.) self-attention layers on CIFAR-10 dataset and rectangular missingness. Lower FID score is better.

Figure 7 shows a comparison of classification accuracies for the Landsat dataset achieved using different ensemble sizes ($N$). As it can be seen from this figure, higher values of $N$ result in improved accuracies, especially for higher missing rates. Also, it can be observed that for N values more than 64 the difference is negligible.

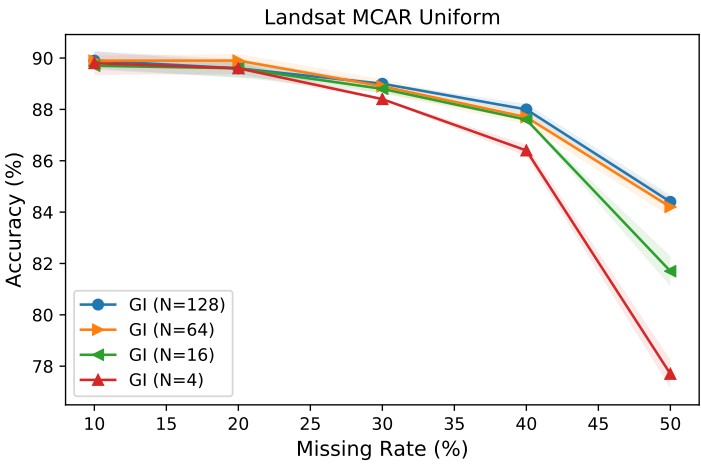

Figure 7: Comparison of classification accuracies achieved with different ensemble size ($N$).

To study the benefits of the suggested stochastic predictor, we conducted experiments comparing GI with its non-stochastic variation (N=1). Here, the CIFAR-10 dataset with the rectangular missing structure and missing rates from 20% to as high as 90% is used. From Table 6 it can be inferred that as the rate of missingness increases, the benefits of the suggested predictor algorithm increase significantly. We hypothesize that at higher rates of missingness, the conditional distribution of missing features becomes multimodal. In such a scenario, the suggested method captures the uncertainties over the target distribution resulting in the predictor to make more reliable class assignments.

Table 6: Comparison of CIFAR-10 accuracies for the stochastic (N=128) and the deterministic (N=1) predictor under rectangular missingness.

| Method | Accuracy at Missing Rate (%) | | | | | |
| --- | --- | --- | --- | --- | --- | --- |
| | 20% | 40% | 60% | 70% | 80% | 90% |
| GI (N=128) | 84.0 | 76.9 | 66.1 | 59.1 | 46.0 | 32.1 |
| GI (N=1) | 83.6 | 75.7 | 65.1 | 56.7 | 42.8 | 29.4 |
| % difference (normalized) | 0.5 | 1.6 | 1.5 | 4.1 | 6.9 | 8.4 |

# E   VISUAL COMPARISON

Table 7 and 8 provide a visual comparison of GI, MisGAN, and GAIN. For each missingness structure, we compare the best two imputation methods based on FID scores in Figure 2 i.e., GI versus MisGAN for rectangular missingness and GI versus GAIN for uniform missingness. From Table 7 it can be seen that GI is more capable in the reconstruction of fine details such as horse legs, car wheels, or plane wings. Regarding the results provided in Table 8, GI imputed samples are generally sharper and more realistic, which is consistent with our hypothesis about the drawbacks of the MSE term in the GAIN objective function.

Table 7: Visual comparison of GI and MisGAN for rectangular missingness. In this visualization, we compare the best two methods for the rectangular missing structure i.e., GI and MisGAN.

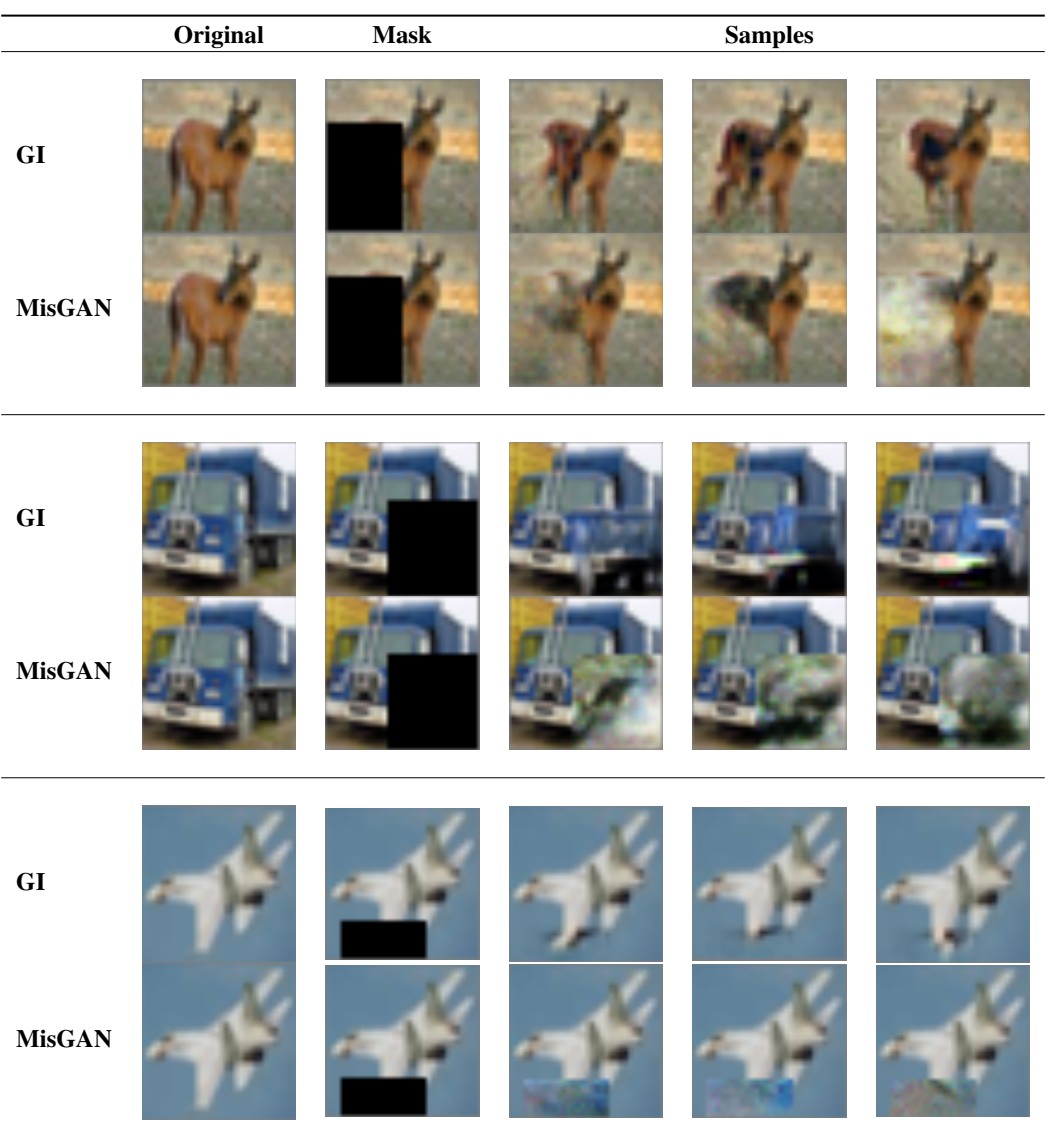

Table 8: Visual comparison of GI and GAIN for uniform missingness. In this visualization, we compare the best two methods for the uniform missing structure i.e., GI and GAIN.

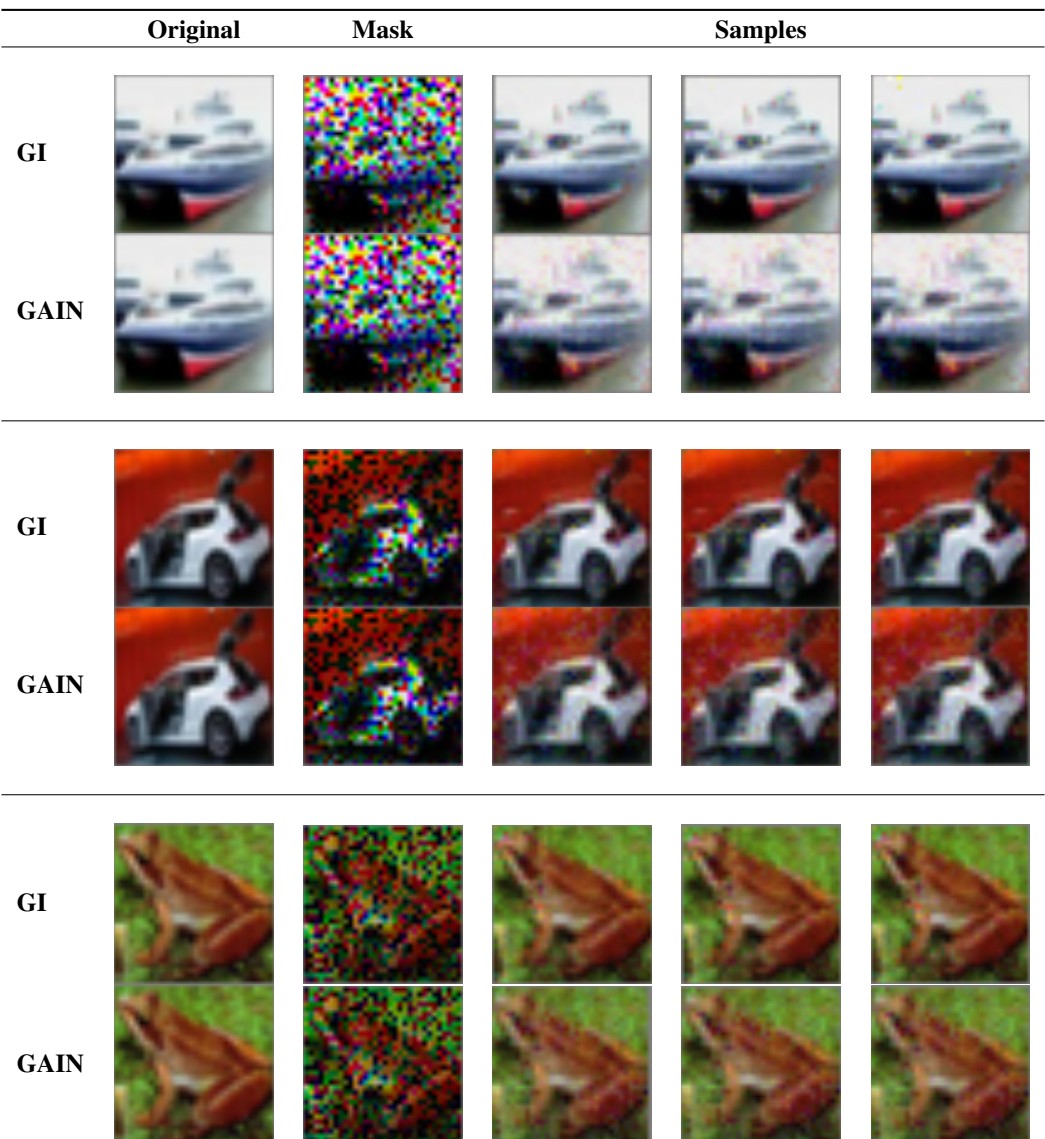

# F ANALYSIS OF THE RMSE MEASURE

Table 9 presents the comparison of different imputation methods using the RMSE measure on CIFAR-10 for different missing structures and rates. Generally, RMSE values for the uniform missing structure are lower than their rectangular counterparts. It is consistent with our intuition that imputing uniform missingness is most similar to denoising problems where the RMSE measure is frequently used. Additionally, comparing the performance of different imputation methods using the FID measure (Section 4.4) does not demonstrate a clear correlation to results shown in Table 9. Nonetheless, it is well-known that the FID measure is more suited to measuring the performance of generated images from the underlying distribution (Heusel et al., 2017).

Similarly, in Table 10, we provide RMSE values corresponding to experiments on the tabular datasets. Here, GAIN and DAE provide very similar results that are generally better than GI or MisGAN. This signifies our hypothesis that the MSE loss term may skew generated samples toward the mean of the distribution, resulting in better RMSE values but not necessarily higher final classification accuracies (see Table 2).

Table 9: Comparison of imputation RMSE values for CIFAR-10 at different missing structures and rates.

| | RMSE at Missing Rate (%) | | | | | |
| | **MCAR Uniform** | | | **MCAR Rect.** | | |
| **Method** | **20%** | **40%** | **60%** | **20%** | **40%** | **60%** |
| GI | 0.026 (±0.003) | 0.057 (±0.008) | 0.090 (±0.006) | 0.097 (±0.02) | 0.148 (±0.001) | 0.660 (±0.010) |
| MisGAN | 0.079 (±0.001) | 0.161 (±0.001) | 0.257 (±0.002) | 0.106 (±0.005) | 0.158 (±0.004) | 0.250 (±0.001) |
| GAIN | 0.027 (±0.003) | 0.045 (±0.001) | 0.072 (±0.005) | 0.340 (±0.047) | 0.511 (±0.001) | 0.660 (±0.010) |
| DAE | 0.036 (±0.001) | 0.075 (±0.002) | 0.121 (±0.005) | 0.116 (±0.007) | 0.160 (±0.001) | 0.233 (±0.029) |

Table 10: Comparison of imputation RMSE values for Landsat, MIT-BIH, and Diabetes datasets at different missing rates.

| | | RMSE at Missing Rate (%) | | | |
| **Dataset** | **Method** | **10%** | **20%** | **30%** | **40%** |
| **Landsat** (Dua & Graff, 2017) | GI | 0.040 (±0.005) | 0.067 (±0.007) | 0.076 (±0.020) | 0.136 (±0.002) |
| | MisGAN | 0.068 (±0.001) | 0.096 (±0.001) | 0.118 (±0.001) | 0.136 (±0.001) |
| | GAIN | 0.018 (±0.001) | 0.024 (±0.001) | 0.030 (±0.001) | 0.037 (±0.001) |
| | DAE | 0.020 (±0.001) | 0.031 (±0.001) | 0.041 (±0.001) | 0.052 (±0.001) |
| **MIT-BIH** (Moody & Mark, 2001) | GI | 0.038 (±0.001) | 0.060 (±0.004) | 0.071 (±0.002) | 0.095 (±0.002) |
| | MisGAN | 0.073 (±0.007) | 0.092 (±0.002) | 0.115 (±0.003) | 0.111 (±0.001) |
| | GAIN | 0.032 (±0.008) | 0.046 (±0.001) | 0.055 (±0.004) | 0.067 (±0.007) |
| | DAE | 0.029 (±0.001) | 0.048 (±0.008) | 0.061 (±0.009) | 0.068 (±0.003) |
| **Diabetes** (Kachuee et al., 2019) | GI | 0.080 (±0.002) | 0.118 (±0.008) | 0.149 (±0.020) | 0.189 (±0.009) |
| | MisGAN | 0.082 (±0.004) | 0.111 (±0.002) | 0.133 (±0.001) | 0.151 (±0.001) |
| | GAIN | 0.064 (±0.001) | 0.092 (±0.001) | 0.119 (±0.001) | 0.140 (±0.001) |
| | DAE | 0.065 (±0.001) | 0.093 (±0.001) | 0.118 (±0.001) | 0.143 (±0.001) |

## G  IMPACT OF TRAINING NOISE

Addition of noise to input vectors often serves as an input augmentation and results in improved generalization accuracies. In order to verify that the improved GI performance is not merely due to the introduction of noise in the suggested architecture, we conducted an experiment by adding different amounts of Gaussian noise during the training process for GAIN and GI. Specifically, we compared how the CIFAR-10 test accuracies change at different degrees of training noise for uniform and rectangular missingess structures at the average missing rate of 40%.

According to Table 11, adding small amounts of Gaussian noise (e.g., std=0.0125) improves the generalization under uniform missingness for both GI and GAIN. Even in this case, GI is still outperforming GAIN in terms of final classification performance. It is also interesting to point out that for the case of rectangular missingness adding Gaussian noise results in a consistent reduction in the classification accuracy for both methods.

Table 11: Top-1 CIFAR-10 classification accuracy at 40% missing rate using added training noise.

| | Accuracy (%) | | | |
| --- | --- | --- | --- | --- |
| | MCAR Uniform (40%) | | MCAR Rect. (40%) | |
| Noise STD | GI | GAIN | GI | GAIN |
| 0.0 | 87.1 | 86.0 | **76.9** | 73.6 |
| 0.0125 | **87.3** | 86.3 | 76.8 | 73.3 |
| 0.025 | 86.5 | 86.6 | 76.7 | 73.2 |
| 0.05 | 85.6 | 84.7 | 73.7 | 72.4 |
| 0.1 | 82.0 | 80.6 | 68.7 | 67.0 |

# H    IMPACT OF THE MSE LOSS TERM

In our earlier discussions, we stated that the MSE loss term used in GAIN would bias the distribution of generated samples toward the mean of the distribution. Here, a synthesized dataset is used to illustrate the impact of MSE loss term on the distribution of generated samples. A hyperparameter, $\lambda$, controls the weight of the MSE term in the final objective function. As it can be observed from Figure 8, the higher the $\lambda$ parameter, the lower the variance of the generated samples (i.e., more bias toward the mean of the distribution).

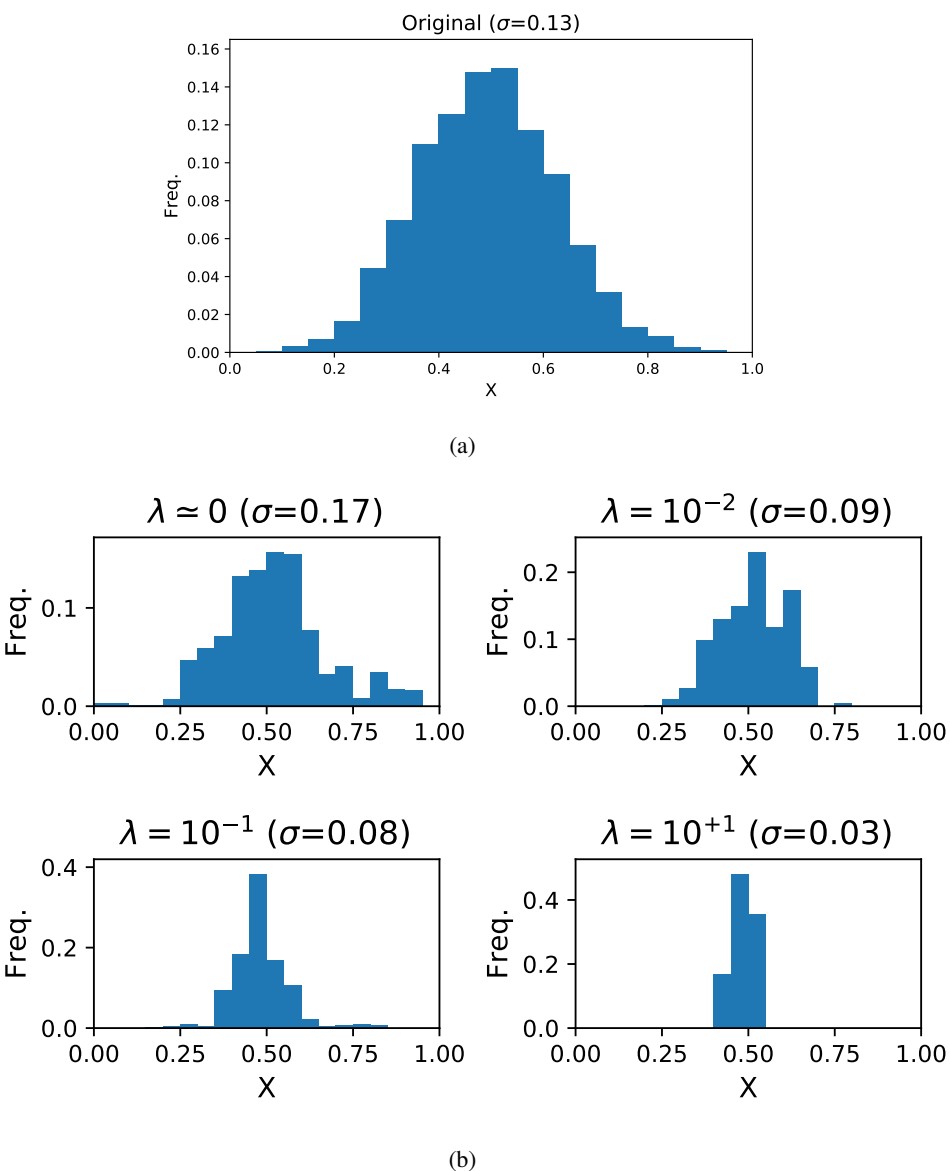

Figure 8: Comparison of generating samples from a Gaussian distribution (a) samples from the original distribution, (b) samples generated using GAIN imputers with different significance of the MSE term (controlled by $\lambda$).

# I   IMPACT OF THE DISCRIMINATOR HINT VECTOR

Yoon et al. (2018) suggested the idea of guiding the discriminator network using a hint mechanism. A hint vector reveals a subset of features that are missing to the discriminator. In Figure 9 and 10 we provide a comparison of learning curves for GI implemented using different hint rates. From Figure 9, using the hint mechanism does not result in any noticeable improvement in the final imputation quality justifying the added complexity. For the case of the rectangular missing structure in Figure 10; however, using the hint vector causes instabilities in the training process. One possible explanation is: providing even a small portion of the mask as a hint, due to the deterministic nature of the rectangular shape it is equivalent to providing region boundaries to the discriminator making it obvious for the discriminator. In GAN training we generally want to have equal competition between the generator and discriminator.

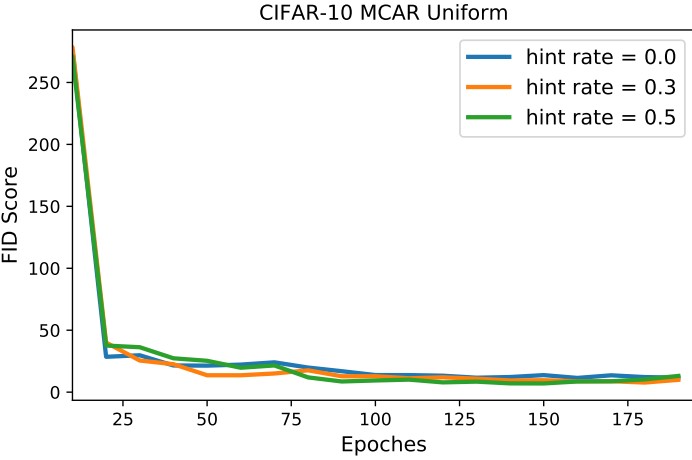

Figure 9: Learning curves for CIFAR-10 with uniform missing structure at different discriminator hint rates.

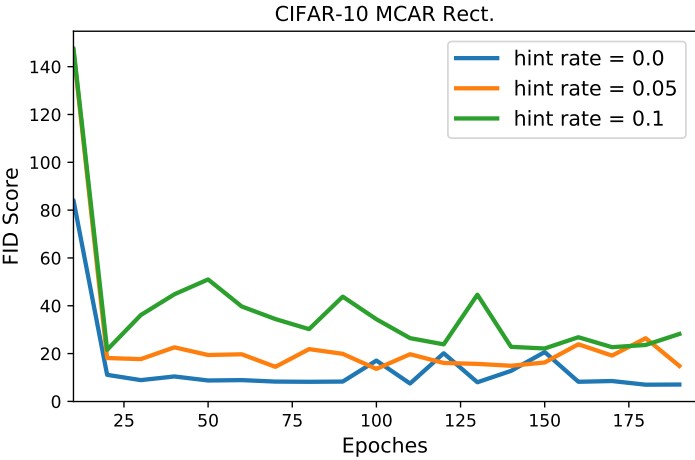

Figure 10: Learning curves for CIFAR-10 with rectangular missing structure at different discriminator hint rates.

