# OpenReview forum: "Generative Imputation and Stochastic Prediction"
_ICLR.cc/2020/Conference — Reject_

### Official Review · AnonReviewer2 · 2019-10-18
**Official Blind Review #2**

**Rating:** 6

**Review:**

The uncertainty of having a missing value is investigated on the prediction by not assigning a single imputed value but N different values generated via an imputer network (based on GAIN). Unlike old-fashion multiple imputation techniques, one predictor is trained on different samples and it induces the uncertainty. The experiments on number of datasets show the proposed predictor is capable of having a fairly better performance.
Overall, this paper raises an interesting point about missing data imputation via generative models, and well-written; however, there are number of concerns:
1-	The predictor is trained on different version of imputed samples (imputed via the generator); this equates to making noisy version of the real samples, where noise is applied to the missing variables. A side effect of this is generalization of the predictor; thus, have you been careful that the improved accuracy is not due to generalization? In other words, if we imposed the generalization via adding gaussian noise to the imputed samples by GAIN for example, would we get improved accuracy too?
2-	Your method known as GI is a modified version of GAIN. You could also use MisGAN, and I am wondering if the results would have been different if generator in MisGAN was used in GI in Figure 2 (b), as MisGAN works better than GAIN.
3-	I am also wondering how the GAIN imputation changes by removing the MSE term? GI discards the MSE term in GAIN, and it changes the distribution of the imputed variables by GAIN. Could you maybe fit a Normal distribution on a chosen imputed variable  (N=128) and visualize how different it is from the distribution of imputed variables with GAIN with MSE term.
4-	In justification for claim 1, it is said “This is equivalent to training models using noisy labels”. This is not accurate: in noisy label prediction, we have one (noisy) y corresponding to each x, in your case there are multiple ys for one sample.
5-	In the implementation details, I cannot fully wrap my head around the part “z vector of size 1/8”; how did you choose this 1/8?


**Experience Assessment:**

I have read many papers in this area.

**Review Assessment: Checking Correctness Of Derivations And Theory:**

I carefully checked the derivations and theory.

**Review Assessment: Checking Correctness Of Experiments:**

I assessed the sensibility of the experiments.

**Review Assessment: Thoroughness In Paper Reading:**

I read the paper thoroughly.

---

> ### Author Response · Authors · 2019-11-13
> **To Reviewer #2 (2/2)**
>
>
> *Comment: “3- I am also wondering how the GAIN imputation changes by removing the MSE term? GI discards the MSE term in GAIN, and it changes the distribution of the imputed variables by GAIN. Could you maybe fit a Normal distribution on a chosen imputed variable  (N=128) and visualize how different it is from the distribution of imputed variables with GAIN with MSE term.”
>
> As suggested, we conducted an experiment to visualize the impact of MSE term on the imputed distribution for a synthesized dataset. The results of this comparison are included in Appendix H of the paper.
> As can be seen from the comparison, the MSE term induces a bias toward the mean and if we increase the hyper-parameter controlling the MSE term  the variance of the generated samples is decreased.
>
> -------------------------------------------
> *Comment: “4- In justification for claim 1, it is said “This is equivalent to training models using noisy labels”. This is not accurate: in noisy label prediction, we have one (noisy) y corresponding to each x, in your case there are multiple ys for one sample.”
>
> We understand the reviewer’s concern; the case of having noisy labels and having imputed samples with different real labels are not exactly equivalent in general. However, if we consider the equivalence in terms of gradient-based training; it can be a valid statement to say that these two cases are very similar.
>
> Note that, during the optimization, we are doing an update based on gradients of loss terms that come from different versions of an imputed sample in which a few cases have a different assigned label as the correct underlying label. Intuitively, we are using misleading gradient terms as it happens in gradient-based learning using noisy labels. If we consider the average impact on gradients for batches of samples rather than the individual cases, the overall impact on the training would be very similar.
>
> In order to address the reviewer's concern, we revised the paper to be more precise about this and omitted the term “equivalent” (see the last paragraph of Claim 1 in Section 3.3).
>
> -------------------------------------------
> *Comment: “5- In the implementation details, I cannot fully wrap my head around the part “z vector of size 1/8”; how did you choose this 1/8?”
>
> We use the z vector as the source of randomness for the generator network. A z vector of very small size would not be effective enough and may result in deterministic imputations given a set of observed features. On the other hand, a very large z vector would unnecessarily increase the size of the input layer for the generator network causing training issues in practice.
>
> Regarding the specific choice of 1/8, we conducted initial experiments using different sizes for the z vector. We found that the final performance is not too sensitive to the choice of z size (as long as it is not relatively small or large). In this paper, we suggest the 1/8 ratio as it is a reasonable choice that does not add a huge overhead and at the same time is strong enough to induce randomness to the generator. For instance, the original implementation of GAN on MNIST (Goodfellow et al. 2014) uses a z of size 100 for inputs of size 784, roughly the same ratio.

---

> > ### Comment · AnonReviewer2 · 2019-11-15
> > **Thanks for the responses.**
> >
> > Thanks you for adding the appendices.
> > With regards to comment #2, do you have any comment/speculation why MisGAN(imputer)+GI(predictor) underperfoms GI(which is based on GAIN)?

---

> > > ### Author Response · Authors · 2019-11-15
> > > **Re: Thanks for the responses**
> > >
> > > Regarding the reason why MisGAN(imputer)+GI(predictor) still underperforms GI, we think the main reason is GI imputer is better than MisGAN imputer (See Figure 2 where GI performs the best in terms of FID score). Since the only difference is in the imputer, using a better imputer is expected to result in better performance.

---

> ### Author Response · Authors · 2019-11-13
> **To Reviewer #2 (1/2)**
>
>
> Thank you for reviewing the manuscript and helpful comments. Please find a point-to-point response to your comments in the following.
>
> -------------------------------------------
> *Comment: “1- The predictor is trained on different version of imputed samples (imputed via the generator); this equates to making noisy version of the real samples, where noise is applied to the missing variables. A side effect of this is generalization of the predictor; thus, have you been careful that the improved accuracy is not due to generalization? In other words, if we imposed the generalization via adding gaussian noise to the imputed samples by GAIN for example, would we get improved accuracy too?”
>
> Thank you for raising this concern. Adding Gaussian noise often serves as a data augmentation and may result in improved generalization.
>
> In order to address this concern, we conducted an experiment by adding different amounts of Gaussian noise during the training process for GAIN and GI. Then we compared how the accuracies change at different rates of training noise. As we thought it might also be interesting to our readers, we included this comparison in Appendix G of the paper.
>
> In summary, adding small amounts of Gaussian noise (e.g., std=0.0125) improves the generalization under uniform missingness for both GI and GAIN. Even in this case, GI is still outperforming GAIN in terms of final classification performance. Therefore, the improved accuracy cannot be solely attributed to the noise generated during the stochastic process.
> It is also interesting to point out that for the case of rectangular missingness, adding Gaussian noise results in a consistent reduction in the classification accuracy for both methods.
>
> -------------------------------------------
> *Comment: “2- Your method known as GI is a modified version of GAIN. You could also use MisGAN, and I am wondering if the results would have been different if generator in MisGAN was used in GI in Figure 2 (b), as MisGAN works better than GAIN.”
>
> We conducted experiments using MisGAN as imputer block for GI under different degrees of rectangular missingness. Here is a comparison of GI, MisGAN, and GI(predictor)+MisGAN(imputer):
>
>
>                           |   MCAR Rect. Missing Rate
>                           |   20%   |   40%     |   60%   |
>  -----------------------------------------------------------
> GI                      |  84.0%  |  76.9%  |   66.1% |
> MisGAN           |  82.9%  |  75.6%  |   65.0% |
> GI+MisGAN     |  83.5%  |  76.5%  |   65.3% |
>
>
> As it can be seen from these results, using MisGAN(imputer)+GI(predictor) performs better than MisGAN, while it is still worse than using the suggested GI imputer and predictor combination.
> Note that from Fig.2(b), GI is the better imputer while MisGAN trails  below with a small margin. It is consistent with the results presented above as GI is still using a better imputer compared to MisGAN+GI.
> We decided not to include this result in the paper at this point; though we would be happy to add this table if the reviewer thinks it would be helpful.

---

### Official Review · AnonReviewer1 · 2019-10-24
**Official Blind Review #1**

**Rating:** 6

**Review:**

This is a nice piece of incremental work on top of previously published GAN imputation methods. It seems to work well in the limited evaluation and is at least claimed to be easier to use for practitioners. This paper could benefit tremendously from both better evaluation and discussion. The paper would be much clearer if GI contextualized itself relative to GAIN on the one hand (which is the most similar GAN method) and multiple imputation on the other hand (of which this is almost, but not quite, an instance of).


Suggestions for improving the introduction & discussion:
* The purpose of this paper is to model uncertainties about missing values — you really should say more about probabilistic methods than "A few exceptions exist such as Bayesian models”.  At least give some motivation for why certain imputations problems couldn’t be feasibly solved by modeling the missing values in a probabilistic programming framework.
* Other GAN methods for imputation (GAIN and MisGAN) are dismissed as "often very complicated to be applied in practical setups by practitioners”. Given that the described method resembles GAIN, is it really much simpler? If so, can you be more specific when characterizing related work?
* "This is different from approaches such as multiple imputation where several predictors are trained on different imputed versions of a dataset.” — the main difference between this approach and MI is that you’re interleaving imputation and training a downstream model. Emphasize this earlier on, since it will make the whole technique easier to understand.

Suggestions for improving the evaluation:
* You’re imputing missing rectangles from an image dataset — please show us the resulting images. I would greatly prefer this to Section 4.5 — which is a very low sample size, low dimensionality example and it’s really unclear how well it generalizes to real data.
* "We also considered using root means squared error (RMSE); however, we decided not to use this measure as we observed an inconsistent behavior using RMSE in our comparisons as RMSE favors methods that show less variance rather than realistic and sharp samples from the distribution.” — I think this is a mistake, likely motivated by the proposed method doing worse under the RMSE metric. Show us several relevant metrics and then discuss their tradeoffs afterward.
* "We run each experiment multiple times (at least 4)” — please report how often each experiment was run, even better if you standardize this number.
* For Table 2, please provide accuracy without missing values as a baseline.
* Add MICE or some other “standard” imputation method as a baseline.

Suggestions for improving readability:
* Many sentences start with “in this” (e.g. “in this case”, “in this setting”, &c). Sometimes these sentences even co-occur within the same paragraph. Try to switch up the phrasing and move away from repetition.
* Not a complete sentence: "For instance, jointly training multiple generator/discriminator networks, tuning objective functions with multiple hyper-parameters, etc."

Update: I think the latest draft of the paper is a big improvement, the inclusion of a "classical" baseline, improved language and additional appendices are all welcome. I'm leaving the rating as a "weak accept" since the paper still feels rough and could use additional editing/streamlining.

**Experience Assessment:**

I have published one or two papers in this area.

**Review Assessment: Checking Correctness Of Derivations And Theory:**

I assessed the sensibility of the derivations and theory.

**Review Assessment: Checking Correctness Of Experiments:**

I assessed the sensibility of the experiments.

**Review Assessment: Thoroughness In Paper Reading:**

I read the paper at least twice and used my best judgement in assessing the paper.

---

> ### Author Response · Authors · 2019-11-13
> **To Reviewer #1 (2/2)**
>
>
> * Comment: “"We also considered using root means squared error (RMSE); however, we decided not to use this measure as we observed an inconsistent behavior using RMSE in our comparisons as RMSE favors methods that show less variance rather than realistic and sharp samples from the distribution.” — I think this is a mistake, likely motivated by the proposed method doing worse under the RMSE metric. Show us several relevant metrics and then discuss their tradeoffs afterward.”
>
> In appendix F of the revised version, we report RMSE values for all experiments. We also provided a discussion about how RMSE values compare to FID values as well as how different imputation methods perform based on the RMSE measure.
>
> -------------------------------------------
> * Comment: “"We run each experiment multiple times (at least 4)” — please report how often each experiment was run, even better if you standardize this number.”
>
> To address your comment we revised the sentence to be more precise about the number of runs (first paragraph of Section 4.3):
> “We run each experiment multiple times: 4 times for CIFAR-10 and 8 times for tabular datasets. We report the mean and standard deviation of results for each case.“
>
> -------------------------------------------
> * Comment: “For Table 2, please provide accuracy without missing values as a baseline.”
>
> As suggested, we included baseline accuracies for complete datasets (0% missing rate) as a footnote to Table 2.
>
> -------------------------------------------
> * Comment: “Add MICE or some other “standard” imputation method as a baseline.”
>
> To address your comment, we included MICE imputation results for all tabular datasets. See the revised Table 2.
>
> -------------------------------------------
> Suggestions for improving readability:
>
> * Comment: “Many sentences start with “in this” (e.g. “in this case”, “in this setting”, &c). Sometimes these sentences even co-occur within the same paragraph. Try to switch up the phrasing and move away from repetition.”
>
> As suggested, we revised the paper to be more clear and reduced unnecessary repetitions.
> We modified about 10 instances of “In this” appearing at the beginning of a sentence throughout the paper.
>
>
> -------------------------------------------
> * Comment: “Not a complete sentence: "For instance, jointly training multiple generator/discriminator networks, tuning objective functions with multiple hyper-parameters, etc."
>
> Thank you for pointing this out. In the revised version, we changed the sentence to be more explicit and fixed the grammar issues:
>
> “For instance, Yoon et al. (2018) requires setting hyperparameters to adjust the influence of an MSE loss term as well as the rate of discriminator hint vectors. Also as another example, Li et al. (2019) uses three generators and three discriminators for the final imputer architecture.”

---

> ### Author Response · Authors · 2019-11-13
> **To Reviewer #1 (1/2)**
>
>
> Thank you for reviewing the manuscript and helpful comments. Please find a point-to-point response to your comments in the following.
>
> -------------------------------------------
> * Comment: “The purpose of this paper is to model uncertainties about missing values — you really should say more about probabilistic methods than "A few exceptions exist such as Bayesian models”.  At least give some motivation for why certain imputations problems couldn’t be feasibly solved by modeling the missing values in a probabilistic programming framework.“
>
> Thank you for pointing this out, we agree with the reviewer that probabilistic and Baysian methods deserve a more precise discussion. We revised the paper to include a discussion on Baysian methods and added proper citations accordingly (see the last paragraph of Section 2):
>
> “Note that while certain Bayesian methods such as probabilistic Bayesian networks allow handling of missing values as unobserved variables. However, given an incomplete training dataset and without any known causal structure as a priori, learning such models is a very challenging problem with the complexity of at least NP-complete to learn the network architecture in addition to an iterative EM optimization to learn model parameters (Darwiche, 2009; Neapolitan et al. 2004).”
>
> -------------------------------------------
> * Comment: “Other GAN methods for imputation (GAIN and MisGAN) are dismissed as "often very complicated to be applied in practical setups by practitioners”. Given that the described method resembles GAIN, is it really much simpler? If so, can you be more specific when characterizing related work?”
>
> GAIN is using an additional MSE loss term as well as a hint vector, imposing a two additional hyperparameters that require tuning.
> In the case of MisGAN it is exacerbated further as it has three generators and three discriminators for the final imputer architecture. In contrast, our method has a single generator and discriminator.
>
> To address your comment and to clarify this for our readers, we added an explanation to the revised version (the fourth paragraph of Section 2):
>
> “For instance, Yoon et al. (2018) requires setting hyperparameters to adjust the influence of an MSE loss term as well as the rate of discriminator hint vectors. Also as another example, Li et al. (2019) is using three generators and three discriminators for the final imputer architecture.”
>
> -------------------------------------------
> * Comment: “"This is different from approaches such as multiple imputation where several predictors are trained on different imputed versions of a dataset.” — the main difference between this approach and MI is that you’re interleaving imputation and training a downstream model. Emphasize this earlier on, since it will make the whole technique easier to understand.”
>
> As suggested, we added a discussion on the relationship of this work and MI earlier in the paper. We agree with you that it will help with motivation and understanding of our readers.
>
> The last paragraph of the introduction:
> “This work suggests the idea of training a predictor on different imputed samples to capture the uncertainties over class assignments. Compared to MI, the suggested method interleaves imputation and training a downstream prediction model, enabling to estimate classification uncertainties for each instance.”
>
> -------------------------------------------
> * Comment: “You’re imputing missing rectangles from an image dataset — please show us the resulting images. I would greatly prefer this to Section 4.5 — which is a very low sample size, low dimensionality example and it’s really unclear how well it generalizes to real data.”
>
> Section 4.5 was designed to be a very simple distribution so that we can visually show the generated samples in a 2-D space.
> In order to address your comment, we included a new appendix in the paper that is dedicated to the visual comparison of imputed samples for different missing structures and different imputers. See Appendix E of the revised version.

---

### Official Review · AnonReviewer3 · 2019-10-25
**Official Blind Review #3**

**Rating:** 6

**Review:**

This paper proposes a method to impute missing features using a generative model and train a predictive model on top of imputed dataset to improve classification results. They first train a GAN model where the generator outputs an imputed representation of the input and discriminator is trained to predict if an individual features (such as a pixel) is imputed or not. Given the generator and incomplete sample, they train a predictor using the output of the generator, imputed sample, as input. Their main contribution is using a MC averaging to compute the prediction by repetitively sampling from the noise variable, z, and generating different imputations from generator. They show that the proposed model improves upon the previous SOTA on final classification performance.

Overall the paper is clearly written. But I do feel it is a bit incremental over the GAIN approach. The overall GAN architecture is very similar to GAIN's and although stochastic prediction shows clear improvements it is a bit straightforward. However, I think the uncertainty of the imputations and its effect on the final prediction is interesting. I suggest the authors to extend this part with more detailed analysis.

There are several parts that are confusing/missing in the paper:

- In GAIN, they use a hint vector as an input to the discriminator. They show that without the hint vector, there is no unique solution (this is shown without the MSE loss). The authors do not use this vector in their approach (as in Figure 1) and it is not clear to me if it causes any instabilities or if multiple experiments yield similar results or if the stochastic prediction benefits from this.
- On what type of examples GI is more accurate than other models? Since stochastic prediction is the main difference from GAIN, is this related to the multi-modality of the noisy examples?
- Can you explain the difference between the results in Figure-7 and Table-2? Results between the two mismatch.
- I think the statement in the first paragraph in Section 4.4 that "MSE loss term would act as a denoising loss smoothing noisy missing pixels" could be misleading. MSE is used with mask in GAIN, hence it only applies to the observed features during training. Its effect on smoothing noisy missing pixels is not clear.


I think the paper would benefit if the authors could explain/show:
- Increasing the missing rate would also increase the possibility that the ground truth be a more multi-modal distribution. Especially in rectangular generation part where it can remove a complete object. Does stochastic averaging benefit more in this case?

**Experience Assessment:**

I have read many papers in this area.

**Review Assessment: Checking Correctness Of Derivations And Theory:**

N/A

**Review Assessment: Checking Correctness Of Experiments:**

I carefully checked the experiments.

**Review Assessment: Thoroughness In Paper Reading:**

I read the paper thoroughly.

---

> ### Author Response · Authors · 2019-11-13
> **To Reviewer #3 (2/2)**
>
>
> * Comment: “Can you explain the difference between the results in Figure-7 and Table-2? Results between the two mismatch.”
>
> Thank you for pointing out the inconsistency. Table 2 provides final classification accuracies for Landsat. To generate results related Figure 7, we used less number of training epochs and did not use the optimal hyper-parameter settings. To address this issue, we rerun the ablation experiments and updated Figure 7 in the revised version to use a similar setup as used for Table 2.
>
> -------------------------------------------
> * Comment: “I think the statement in the first paragraph in Section 4.4 that "MSE loss term would act as a denoising loss smoothing noisy missing pixels" could be misleading. MSE is used with mask in GAIN, hence it only applies to the observed features during training. Its effect on smoothing noisy missing pixels is not clear.”
>
> We understand the reviewer’s concern that the MSE loss is not exactly similar to what we have for denoising autoencoders. However, based on our experiments, its overall impact is very similar to autoencoders. Note that the generator network is partially receiving an autoencoder loss term which is well-known to cause over-smoothing.
> To address the comment and prevent misleading our readers, we revised the sentence to be more scientifically accurate (first paragraph in Section 4.4):
> “One possible explanation for this behavior might be the fact that GAIN has an MSE loss term acting similar to an autoencoder loss smoothing noisy missing pixels.”
>
> -------------------------------------------
> * Comment: “I think the paper would benefit if the authors could explain/show:
> - Increasing the missing rate would also increase the possibility that the ground truth be a more multi-modal distribution. Especially in rectangular generation part where it can remove a complete object. Does stochastic averaging benefit more in this case?”
>
> As suggested by the reviewer, we conducted a new experiment to show the benefit of stochastic averaging over simple classification for rectangular missingness ranging from 20% to 90% (see Table 6 in Appendix D).
>
> Last paragraph of Appendix D:
> “From Table 6, it can be inferred that as the rate of missingness increases, the benefits of the suggested predictor algorithm increase significantly. We hypothesize that at higher rates of missingness, the conditional distribution of missing features becomes multimodal. In such a scenario, the suggested method captures the uncertainties over the target distribution resulting in the predictor to make more reliable class assignments.”

---

> ### Author Response · Authors · 2019-11-13
> **To Reviewer #3 (1/2)**
>
> Thank you for reviewing the manuscript and helpful comments. Please find a point-to-point response to your comments in the following.
>
> -------------------------------------------
> * Comment: “In GAIN, they use a hint vector as an input to the discriminator. They show that without the hint vector, there is no unique solution (this is shown without the MSE loss). The authors do not use this vector in their approach (as in Figure 1) and it is not clear to me if it causes any instabilities or if multiple experiments yield similar results or if the stochastic prediction benefits from this.”
>
> To answer your comment, we included a new appendix to the revised paper studying the impact of the discriminator hint vector (See Appendix I).
>
> In summary:
> - For the uniform missing structure, using the hint mechanism does not result in any noticeable improvement in the final imputation quality.
> - For the rectangular missing structure, however, using the hint vector causes instabilities in the training process. One possible explanation is: providing even a small portion of the mask as a hint, due to the deterministic nature of the rectangular shape  it is equivalent to providing region boundaries to the discriminator making it obvious for the discriminator. In GAN training we generally want to have equal competition between the generator and discriminator.
>
> Based on these observations, we decided not to use hint vector in our final architectures.
>
> -------------------------------------------
> * Comment: “On what type of examples GI is more accurate than other models? Since stochastic prediction is the main difference from GAIN, is this related to the multi-modality of the noisy examples?”
>
>
> We believe that this work (GI) outperforms GAIN based on two factors:
>
> First, in contrast to GAIN, the GI imputer objective function does not have an MSE loss term. We hypothesize that using an MSE loss would lead to generated samples to be biased toward the mean of the underlying data generation distribution. To support this hypothesis, we included a new appendix to the revised version to show the impact of the additional loss term on the generated samples for a synthesized dataset (see Appendix H).
>
> Second, given that GI is able to generate sharp samples from a multimodal distribution, the stochastic predictor would help model the distribution of target class assignments based on an ensemble of samples (128 in this paper) rather than an over-confident assignment using a single imputed sample. Section 4.5 of the paper uses a multi-modal synthesized dataset to show this visually.

---

### Author Response · Authors · 2019-11-13
**Summary of changes**

We thank the reviewers for constructive comments and suggestions. We believe that the suggested revisions enhanced the scientific quality of the manuscript significantly.

The summary of revisions is as follows:

- We included five new appendices to the revised version, studying different questions raised by our reviewers:
Appendix E: “Visual Comparison”,
Appendix F: “Analysis of the RMSE Measure”,
Appendix G: “Impact of Training Noise”,
Appendix H: “Impact of the MSE Loss Term”,
Appendix I: “Impact of the Discriminator Hint Vector”.

- In the revised version, we report MICE imputation results as a standard baseline for tabular datasets.

- We included/revised discussions of related work and how the proposed method is compared to them. Also, we revised Claim 1 to be more precise on how learning from imputed samples relates to label noise.

- As suggested by our reviewers, we fixed a few grammar/wording issues to enhance the readability of the paper.

---

### Decision · Program_Chairs · 2019-12-19

**Decision:**

Reject

**Comment:**

The paper proposes a method that does uncertainty modeling over missing data imputation using a framework based on generative adversarial network. While the method shows some empirical improvements over the baselines, reviewers have found the work incremental in terms of technical novelty over the existing GAIN approach which renders it slightly below the acceptance threshold for the main conference, particularly in case of space constraints in the program.